# Estimation of the Cadmium Nephrotoxicity Threshold from Loss of Glomerular Filtration Rate and Albuminuria

**DOI:** 10.3390/toxics11090755

**Published:** 2023-09-06

**Authors:** Soisungwan Satarug, David A. Vesey, Tanaporn Khamphaya, Phisit Pouyfung, Glenda C. Gobe, Supabhorn Yimthiang

**Affiliations:** 1The Centre for Kidney Disease Research, Translational Research Institute, Brisbane 4102, Australia; david.vesey@health.qld.gov.au (D.A.V.); g.gobe@uq.edu.au (G.C.G.); 2Department of Kidney and Transplant Services, Princess Alexandra Hospital, Brisbane 4102, Australia; 3Occupational Health and Safety, School of Public Health, Walailak University, Nakhon Si Thammarat 80160, Thailand; tanaporn.kh@mail.wu.ac.th (T.K.); phisit.po@mail.wu.ac.th (P.P.); ksupapor@mail.wu.ac.th (S.Y.); 4School of Biomedical Sciences, The University of Queensland, Brisbane 4072, Australia; 5NHMRC Centre of Research Excellence for CKD QLD, UQ Health Sciences, Royal Brisbane and Women’s Hospital, Brisbane 4029, Australia

**Keywords:** albuminuria, cadmium, chronic kidney disease, estimated GFR, hypertension, smoking

## Abstract

Cadmium (Cd) is a pervasive, toxic environmental pollutant that preferentially accumulates in the tubular epithelium of the kidney. Current evidence suggests that the cumulative burden of Cd here leads to the progressive loss of the glomerular filtration rate (GFR). In this study, we have quantified changes in estimated GFR (eGFR) and albumin excretion (E_alb_) according to the levels of blood Cd ([Cd]_b_) and excretion of Cd (E_Cd_) after adjustment for confounders. E_Cd_ and E_alb_ were normalized to creatinine clearance (C_cr_) as E_Cd_/C_cr_ and E_alb_/C_cr_. Among 482 residents of Cd-polluted and non-polluted regions of Thailand, 8.1% had low eGFR and 16.9% had albuminuria (E_alb_/C_cr_) × 100 ≥ 20 mg/L filtrate. In the low Cd burden group, (E_Cd_/C_cr_) × 100 < 1.44 µg/L filtrate, eGFR did not correlate with E_Cd_/C_cr_ (β = 0.007) while an inverse association with E_Cd_/C_cr_ was found in the medium (β = −0.230) and high burden groups (β = −0.349). Prevalence odds ratios (POR) for low eGFR were increased in the medium (POR 8.26) and high Cd burden groups (POR 3.64). Also, eGFR explained a significant proportion of E_alb_/C_cr_ variation among those with middle (η^2^ 0.093) and high [Cd]_b_ tertiles (η^2^ 0.132) but did not with low tertiles (η^2^ 0.001). With an adjustment of eGFR, age and BMI, the POR values for albuminuria were increased in the middle (POR 2.36) and high [Cd]_b_ tertiles (POR 2.74) and those with diabetes (POR 6.02) and hypertension (2.05). These data indicate that (E_Cd_/C_cr_) × 100 of 1.44 µg/L filtrate (0.01–0.02 µg/g creatinine) may serve as a Cd threshold level based on which protective exposure guidelines should be formulated.

## 1. Introduction

Chronic kidney disease (CKD) is a progressive disease with high morbidity and mortality, affecting 8% to 16% of the world’s population [1,2,3]. CKD is diagnosed when the estimated glomerular filtration rate (eGFR) falls below 60 mL/min/1.73 m^2^, termed low eGFR, or albuminuria, defined as albumin to creatinine ratios (ACR) ≥ 30 mg/g creatinine in women and ≥ 20 mg/g creatinine in men that persists for at least 3 months [1,2]. Environmental exposure to the toxic metal pollutant cadmium (Cd) is linked to an increased risk of both low eGFR and albuminuria. Typically, studies of CKD associated with environmental Cd exposure employed levels of blood Cd ([Cd]_b_) and excretion of Cd (E_Cd_), normalized to creatinine excretion (E_Cd_/E_cr_) as indicators of exposure [3,4]. Notably, however, these Cd exposure indicators have been found to be also associated with an increased risks of diabetes [5,6,7,8,9,10] and hypertension [11,12,13,14,15,16]; both of which are major risk factors for CKD. Hypertension is known to be a cause and result of CKD [17,18]. Furthermore, in a Dutch prospective cohort study, smoking, which is a source of Cd exposure, promoted kidney failure, evident from a reduction in eGFR to below 15 mL/min/1.73 m^2^ [19]. It is imperative that these covariates are adequately adjusted for the estimation of risks of GFR loss and albuminuria due to renal Cd accumulation.

The dose–response relationship between Cd exposure and adverse kidney outcomes measured by a reduction in eGFR and albuminuria is not well defined. In comparison, numerous dose-effect studies employed tubular damage and tubulopathy reflected by tubular proteinuria as signs of the nephrotoxicity due to Cd accumulation. However, current evidence suggests that a sustained decrease in eGFR after Cd exposure is more suitable than the tubulopathy endpoint, especially for the purpose of determining protective exposure guidelines.

Also, current evidence suggests that excreted Cd (E_Cd_) is related to tubular cell injury and death as a result of the cumulative burden of Cd [20]. The excreted Cd emanates from tubular epithelial cells complexed with metallothionein (MT) as CdMT [21], the storage form of Cd. Consequently, it is logical to normalize E_Cd_ to creatinine clearance (C_cr_) rather than creatinine excretion (E_cr_) because C_cr_ is a surrogate for GFR, which is the measurable analogue of nephron number [3,20,22].

The present study aims to explore if there is a potential toxicity threshold level of Cd accumulation in kidneys. To achieve this aim, changes in eGFR and albumin excretion (E_alb_) together with the risks of low eGFR and albuminuria were quantified in relation to both E_Cd_ and [Cd]_b_. The confounding impacts of smoking, diabetes and hypertension were also evaluated. We collected data from women and men who resided in Cd-polluted and non-polluted regions of Thailand to obtain a wide range of exposure levels pivotal to the a dose–response analysis.

To depict an amount of Cd and albumin excreted per volume of filtrate, known also as primary urine, E_Cd_ and E_alb_ were normalized to C_cr_ as E_Cd_/C_cr_ and E_alb_/C_cr_, respectively [23]. This C_cr_-normalization corrects for differences in urine flow rate (dilution) and the number of functioning nephrons among cohort participants; they are unaffected by creatinine excretion (E_cr_) [23]. Therefore, (E_alb_/C_cr_) × 100 ≥ 20 mg/L filtrate identifies albuminuria in both men and women. In comparison, when E_alb_ is adjusted to creatinine excretion (E_cr_) as albumin to creatinine ratios (ACR), cut-off values for albuminuria in women and men differ because of universally lower E_cr_ values in women than in men.

## 2. Materials and Methods

### 2.1. Cohort Participants

To establish a clear dose–response relationship, a population exposed to a wide range of Cd doses is required. Therefore, we selected subjects from two population-based cross-sectional studies undertaken in a Cd-contaminated area of the Mae Sot District, Tak Province [24] and a non-contaminated location in the Nakhon-Si-Thammarat Province [25] of Thailand. More females (*n* = 354) were recruited to the present study than males (*n* = 118), given that an increase in mortality from kidney disease was found especially in women in a prospective cohort study of residents of a Cd-polluted area of Japan [26].

The data from a nationwide survey of Cd levels in soils and food crops indicated that environmental exposure to Cd in Nakhon Si Thammarat was low [27]. In comparison, the Cd concentration of the paddy soil samples from the Mae Sot district exceeded the standard of 0.15 mg/kg and the rice samples collected from household storage contained four times the amount of the permissible Cd level of 0.1 mg/kg [28]. An independent health survey revealed the prevalence of low eGFR among those resided in the Cd-contaminated area of the Mae Sot District to be 16.1% [29].

The study protocol for the Mae Sot group was approved by the Institutional Ethical Committees of Chiang Mai University and the Mae Sot Hospital (Approval No. 142/2544, 5 October 2001) [24]. The Office of the Human Research Ethics Committee, Walailak University of Thailand, approved the study protocol for the Nakhon Si Thammarat group (Approval number WUEC-20-132-01, 28 May 2020) [25].

Prior to participation, all participants provided informed consent. They had lived at their current addresses for at least 30 years. Exclusion criteria were pregnancy, breast-feeding, a history of metal work and a hospital record or physician’s diagnosis of an advanced chronic disease. Hypertension was defined as systolic blood pressure ≥ 140 mmHg, diastolic blood pressure ≥ 90 mmHg [30], a physician’s diagnosis or prescription of anti-hypertensive medications. Diabetes was diagnosed when fasting plasma glucose levels ≥ 126 mg/dL (https://www.cdc.gov/diabetes/basics/getting-tested.html) (accessed on 12 August 2023) or a physician’s prescription of anti-diabetic medications.

### 2.2. Assessment of Cadmium Exposure and Adverse Effects

Assessment of Cd exposure was based on a one-time measurement of blood Cd concentration ([Cd]_b_) and the urinary excretion of Cd (E_Cd_). Kidney functional assessment was based on E_alb_ and eGFR. For these measurements, samples of urine and whole blood were collected after overnight fast. Blood samples were collected within 3 h of urine collection. Aliquots of blood and urine samples were stored at −80 °C for later analysis.

Levels of Cd in urine and blood ([Cd]_u_ and [Cd]_b_) were quantified by graphite furnace atomic absorption spectrometry with the Zeeman effect background correction system. Multielement standards (Merck KGaA, Darmstadt, Germany) were used for instrument calibration. The quality control and quality assurance of Cd quantitation were accomplished by simultaneous analysis of blood control samples (ClinChek, Munich, Germany) and the reference urine metal control levels 1, 2 and 3 (Lyphocheck, Bio-Rad, Hercules, CA, USA).

The limit of detection (LOD) for Cd in blood or urine, defined as 3 times the standard deviation of at least 10 blank sample measurements, was 0.3 µg/L for [Cd]_b_ and 0.1 µg/L for [Cd]_u_. The sample blanks as reference standards for urine and blood were included in the assay together with samples of urine or blood from subjects. Deionized water was used to zero an instrument. The coefficient of variation in Cd in the reference urine and blood were within acceptable clinical chemistry standards. When a sample contained Cd below its LOD, the Cd concentration assigned was the LOD value divided by the square root of 2 [31].

Urinary and plasma creatinine concentrations ([cr]_u_ and [cr]_p_) were measured by the colorimetric method, based on the alkaline-picrate Jaffe’s reaction [32]. Urinary albumin concentration was determined by an immunoturbidimetric method [33,34].

### 2.3. Estimated Glomerular Filtration Rate (eGFR)

In theory, GFR is a measure of nephron function which is the product of the functioning nephron number and mean single nephron GFR [35,36,37]. In practice, GFR is estimated from established chronic kidney disease epidemiology collaboration (CKD-EPI) equations and is reported as estimated GFR (eGFR) [34]. These CKD-EPI equations have been validated with inulin clearance [38].

Male eGFR = 141 × [plasma creatinine/0.9]^Y^ × 0.993^age^, where Y = −0.411 if [cr]_p_ ≤ 0.9 mg/dL and Y = −1.209 if [cr]_p_ > 0.9 mg/dL. Female eGFR = 144 × [plasma creatinine/0.7]^Y^ × 0.993^age^, where Y = −0.329 if [cr]_p_ ≤ 0.7 mg/dL and Y = −1.209 if [cr]_p_ > 0.7 mg/dL. CKD stages 1, 2, 3a, 3b, 4 and 5 corresponded to eGFRs of 90–119, 60–89, 45–59, 30–44, 15–29 and <15 mL/min/1.73 m^2^, respectively. For dichotomized comparison, CKD (abnormal eGFR) was defined eGFR ≤ 60 mL/min/1.73 m^2^.

### 2.4. Normalization of Excretion Rate

Excretion of x, E_x_ was normalized to E_cr_ as [x]_u_/[cr]_u_, where x = Cd or alb; [x]_u_ = urine concentration of x (mass/volume) and [cr]_u_ = urine creatinine concentration (mg/dL). The ratio [x]_u_/[cr]_u_ was expressed in μg/g of creatinine.

E_x_ was normalized to C_cr_ as E_x_/C_cr_ = [x]_u_[cr]_p_/[cr]_u_, where x = Cd or alb; [x]_u_ = urine concentration of x (mass/volume); [cr]_p_ = plasma creatinine concentration (mg/dL) and [cr]_u_ = urine creatinine concentration (mg/dL). E_x_/C_cr_ was expressed as the excretion of x per volume of filtrate [23].

Results obtained with C_cr_-normalized data are shown herein. Results of analogous analyses with E_cr_-normalized data are provided in Appendix A.

### 2.5. Statistical Analysis

Data were analysed with IBM SPSS Statistics 21 (IBM Inc., New York, NY, USA). The Mann–Whitney U test was used to compare two groups and Pearson’s chi-squared test was used to assess differences in percentages. Distribution of the variables was examined for skewness and those showing right skewing were subjected to logarithmic transformation before analysis, where required. Departure from normal distribution of variables was assessed by one sample Kolmogorov–Smirnov test.

A multiple linear regression model analysis was used to identify predictors of eGFR and E_alb_/C_cr_. The multivariable logistic regression analysis was used to determine the prevalence odds ratio (POR) for low eGFR and albuminuria. Univariate analysis of covariance via Bonferroni correction in multiple comparisons was used to obtain covariate adjusted mean E_alb_/C_cr_ and eta square (η^2^) values. For all tests, *p*-values ≤ 0.05 were considered to indicate statistical significance.

## 3. Results

### 3.1. Cohort Participants

Table 1 provides descriptive characteristics of the Thai cohort participants, who were recruited from a low-exposure locality and a high-exposure region due to environmental Cd pollution.

A total of 482 persons (354 women and 118 men), mean age of 51.8 years, were included in this investigation. The respective percentages of smoking and diabetes were 29.7% and 18.3%, while about half (48.5%) had hypertension.

The overall % low eGFR was 8.1% and % abnormal ACR was 15%. The overall % abnormal E_alb_/C_cr_ was 16.9%. Of note, the % abnormal E_alb_ was higher in the low eGFR group than normal eGFR group for both ACR and E_alb_/C_cr_ criteria in both genders. Similarly, the % hypertension and diabetes were higher in the low eGFR than the normal eGFR group.

The mean [Cd]_b_ (range) was 2.60 (0.03–20) µg/L. The mean (E_Cd_/C_cr_) × 100 (range) was 3.20 (0.02–25) µg/L filtrate. The corresponding mean E_Cd_/E_cr_ (range) was 4.05 (0.03–31) µg/g creatinine.

### 3.2. Predictors of eGFR

Table 2 provides the results of the multiple linear regression analysis to define the independent variables that contributed to differences in eGFR.

In an inclusive model, all eight independent variables explained 27.8% of the total variation in eGFR. Age, E_Cd_/C_cr_ and diabetes were the three main influential independent variables. All other five independent variables did not show a significant association with eGFR. In a subgroup analysis, female eGFR was inversely associated with age (β = −0.511), E_Cd_/C_cr_ (β = −0.126) and diabetes (β = −0.119). In comparison, male eGFR was associated with age only (β = −0.472).

In an analogous regression of eGFR with E_Cd_ normalized to creatinine excretion (E_Cd_/E_cr_) (Appendix A), eGFR was inversely associated with age only (β = −0.442). No significant association was seen between eGFR and all other six variables, including E_Cd_/E_cr_.

To investigate the dose–response relationship of Cd exposure and changes in eGFR, we designated the low, middle and high E_Cd_/C_cr_ tertiles to represent low, medium and high Cd burdens, respectively. In women, the cut-off values of (E_Cd_/C_cr_) × 100 for the low, medium and high Cd burdens were ≤1.44, 1.45–3.26, >3.26 µg/L filtrate, respectively. Corresponding cut-off values of (E_Cd_/C_cr_) × 100 in men were ≤1.25, 1.26–3.25, >3.25 µg/L filtrate.

Figure 1 provides scatterplots relating eGFR to Cd excretion rate in women and men and their subgroups according to Cd burden levels, described above.

In scatterplots including all women (Figure 1a) and all men (Figure 1b), a direct relationship between eGFR and E_Cd_/C_cr_ was suggested. In subgroup analysis, a direct relationship of eGFR and E_C_d/C_cr_ was seen in women and men who had low Cd burden (Figure 1c,d). In contrast, eGFR was inversely related to E_Cd_/C_cr_ in women and men of the medium plus high Cd burden groups (Figure 1e,f). These data indicate that an E_Cd_/C_cr_ of 1.44 µg/L filtrate may be a threshold level of Cd accumulation, above which eGFR began to drop.

To confirm or dispute the contrasting eGFR responses below or above E_Cd_/C_cr_ of 1.44 µg/L filtrate, multiple regression analyses of eGFR across Cd burden groups were undertaken. Results are provided in Table 3.

In distinction from a bivariate analysis (Figure 1c,d), the association between eGFR and E_Cd_/C_cr_ was statistically insignificant in the low Cd burden group after adjustment for confounding variables (Table 3). In this low Cd burden group, eGFR was inversely associated with age only (β = −0.455), while its association with E_Cd_/C_cr_ became statistically insignificant (Table 3).

Of interest, there was an inverse relationship between eGFR and E_Cd_/C_cr_ in the medium and high Cd burden groups following a similar adjustment for confounders. All eight independent variables explained, respectively, 32.1% and 25.5% of total eGFR variation in the medium and high Cd burden groups. In the high Cd burden group, eGFR was more closely associated with E_Cd_/C_cr_ (β = −0.349), compared with the medium Cd burden group (β = −0.284). Other predictors of eGFR in the medium Cd burden group were age (β = −0.505) and systolic blood pressure (β = −0.230). Age was the other predictor of eGFR (β = −0.410) in the high Cd burden group.

### 3.3. Logistic Regression Analysis of Low eGFR

Table 4 provides the results of the logistic regression analysis that evaluated the effects of Cd body burden and other six independent variables on the prevalence odds ratios (POR) for low eGFR. POR is defined in Section 2.5.

In all subjects, the POR values for low eGFR increased with age (POR 1.118) and in those with diabetes (POR 3.024), medium Cd burden (POR 8.265) and high Cd burden (POR 3.643). All other four independent variables did not significantly affect the POR for low eGFR.

Among women, the POR values for low eGFR increased with age (POR 1.114) and in the medium Cd burden group (POR 7.204). An increase in the POR for low eGFR in the high Cd burden group did not reach a statistically significant level (POR 3.218, *p* = 0.064).

An effect of Cd exposure on the POR for low eGFR in men could not be evaluated due a small number of men with low eGFR (*n* = 4). However, the POR for low eGFR among men was associated with hypertension (POR 2.478).

### 3.4. Multiple Regression Analysis of Albumin Excretion Rate

We used multiple regression analysis to identify if E_Cd_/C_cr_ and other independent variables predicted rising E_alb_/C_cr_. We included eGFR in regression models to evaluate if functioning nephrons, indicated by eGFR values, had an independent effect on E_alb_/C_cr_. Results are provided in Table 5.

In the low and medium Cd burden groups, neither E_Cd_/C_cr_ nor eGFR showed a significant association with E_alb_/C_cr_. However, diabetes and hypertension predicted E_alb_/C_cr_ in the low Cd burden group, while BMI was a sole predictor of E_alb_/C_cr_ in the medium Cd burden group.

In the high Cd burden group, E_Cd_/C_cr_ and eGFR both were associated with E_alb_/C_cr_. There was an inverse association between eGFR and E_alb_/C_cr_ (β = −0.214), while E_Cd_/C_cr_ showed a direct relationship with E_alb_/C_cr_ (β = 0.173). All other six independent variables did not have a significant effect on E_alb_/C_cr_.

Of note, the relationship between albumin excretion and eGFR was obscure, when E_alb_ was normalized to E_cr_ (E_alb_/E_cr_). As data in Appendix A indicate, E_alb_/E_cr_ did not show a significant inverse association with eGFR (β = −0.092, *p* = 0.078).

We next conducted another multiple regression analysis to evaluate associations of E_Cd_/C_cr_. eGFR and other independent variables on E_alb_/C_cr_ in participants with normal kidney function (eGFR > 60 mL/min/1.73 m^2^) and those with a sign of CKD (eGFR ≤ 60 mL/min/1.73 m^2^). Results are provided in Table 6.

E_alb_/C_cr_ was inversely associated with eGFR (β = −0.188) when all subjects were included in an analysis. In a subgroup analysis, no association was observed between E_alb_/C_cr_ and eGFR in those with normal kidney function; there was a very strong inverse association of these two variables in those with CKD (β = −0.637).

In a subsequent analysis, scatterplots were used together with a univariate analysis to quantify the independent effects of E_Cd_/C_cr_, eGFR, diabetes and hypertension on E_alb_/C_cr_. Covariate adjusted mean E_alb_/C_cr_ values were obtained for various subgroups, as shown in Figure 2.

In the (E_Cd_/C_cr_) ×100 < 1 µg/L filtrate group (Figure 2a), a scatterplot indicated that the correlation of E_alb_/C_cr_ and E_Cd_/C_cr_ was statistically insignificant. However, in an analysis restricted to those with normal eGFR, differences in E_alb_/C_cr_ were observed in smokers and those with diabetes after adjustment for covariates and interactions (Figure 2b).

In the (E_Cd_/C_cr_) × 100 ≥ 1 µg/L filtrate group (Figure 2c), a scatterplot indicated a statistically significant relationship between E_alb_/C_cr_ and E_Cd_/C_cr_. An increment of E_alb_/C_cr_ was observed in smokers in an analysis restricted to those with normal eGFR after adjustment for covariates and interactions (Figure 2d).

### 3.5. Blood Cadmium and eGFR as Predictors of Albuminuria

Table 7 provides the results of the logistic regression analysis that evaluated if [Cd]_b_, like E_Cd_, predicted an increase in risk of abnormally high E_alb,_ defined as (E_alb_/C_cr_) × 100 ≥ 20 µg/L filtrate.

In all subjects, eGFR, diabetes, hypertension and [Cd]_b_ were independent variables showing associations with the POR for albuminuria. Comparing those with [Cd]_b_ < 0.82 µg/L, POR values for albuminuria were increased by 2.4-fold and 2.7-fold in those with [Cd]_b_ of 0.83–2.63 and ≥2.64 µg/L, respectively. For every 1 mL/min/1.73 m^2^ loss of eGFR, the POR for albuminuria rose by 4.3%. There was no effect of Cd on albuminuria in an analysis with Ecr-normalized data (Appendix A).

Among women, there was a [Cd]_b_-related increment of the POR for albuminuria. The POR values for high albuminuria were increased 3.4-fold and 3.8-fold in those with middle and high [Cd]_b_ tertiles, respectively. In addition, eGFR, diabetes and hypertension were independent variables showing associations with the POR for albumin in women. For every 1 mL/min/1.73 m^2^ loss of eGFR, the POR for albuminuria rose by 4.5%.

Among men, [Cd]_b_ did not show an association with POR for albuminuria, while eGFR did. For every 1 mL/min/1.73 m^2^ loss of eGFR, the POR for albuminuria rose by 4.5%, similar to the effect size found in women.

In a subsequent analysis, covariate adjusted mean E_alb_/C_cr_ values were obtained for groups of participants with different eGFR levels (≤60, 61–90 and >90 mL/min/1.73 m^2^) to reveal an effect of GFR loss, reflected by a reduction in eGFR. Results are shown in Figure 3.

In the low [Cd]_b_ tertile group, the relationship between E_alb_/C_cr_ and eGFR was statistically insignificant (Figure 3a). Similarly, the covariate adjusted mean E_alb_/C_cr_ values did not show a significant variation across three eGFR subgroups.

In the middle and high [Cd]_b_ tertile groups, an inverse relationship was observed between E_alb_/C_cr_ and eGFR (Figure 3c,e); an effect of GFR loss on E_alb_/C_cr_ was apparent (Figure 3d,f). In these two [Cd]_b_ tertile groups, the mean E_alb_/C_cr_ value was highest, middle and lowest in those with eGFR ≤ 60, 60–90 and > 90 mL/min/1.73 m^2^, respectively.

## 4. Discussion

Herein, we used the eGFR decline as a primary endpoint in our attempt to define a toxic accumulation level of Cd in the kidney. We focused on this eGFR because attenuation of eGFR decline is employed in clinical trials to evaluate the effects of treatment of CKD [1,2]. Albuminuria was concurrently examined because it has often been associated with Cd exposure, along with low eGFR [4]. Previous studies show that the burden of Cd as µg/g kidney tissue weight increases with age [39,40,41]. They show also that sufficient tubular damage from the Cd accumulation reduces the GFR, leading to nephron atrophy and a decreasing GFR as a result [20,22,42,43]. We hypothesized that this eGFR effect of Cd occurs at a very low kidney burden, producing E_Cd_/E_cr_ below 5.24 µg/g creatinine, which is the current World Health Organization health-based exposure limit, detailed in Section 4.3. It is noteworthy that E_Cd_ itself is indicative of tubular cell injury and death [3,20] and that E_Cd_/E_cr_ of 5.24 µg/g creatinine signifies extensive tubular cell damage and death.

We analysed data from 482 Thai nationals, of which 8.1% had low eGFR and 15% had albuminuria (ACR criterion). Nearly half (48.3%) had hypertension, while 29.7% reported to be smokers and 18.3% had diabetes. There was a 1250-fold difference in E_Cd_/C_cr_ values, ranging from 0.0002 to 0.25 µg/L filtrate (mean 0.0032 µg/L filtrate). There was a 667-fold difference in [Cd]_b_, ranging between 0.03 and 20 µg/L (mean 2.60 µg/L). The wide ranges of these Cd burden and exposure matrices together with sufficiently high numbers of smokers and those with diabetes and hypertension provide an ideal scenario to define with certainty a toxic threshold level of Cd.

Like E_Cd_/C_cr_, E_Cd_/E_cr_ showed a large variation (1033-fold) and the mean E_Cd_/E_cr_ (range) was 4.05 (0.03–31) µg/g creatinine (Table 1). However, in distinction from E_Cd_/C_cr_, there was no significant association between eGFR and E_Cd_/E_cr_ (Appendix A). Consequently, a dose–response analysis was not possibly determined. Also, when E_alb_ was adjusted to creatinine excretion (E_cr_) as E_alb_/E_cr_ or ACR, an inverse relationship of ACR and eGFR was obscure (Appendix A) and there was no effect of Cd on POR for albuminuria (Appendix A).

### 4.1. Effects of Cadmium on eGFR

A direct relationship between eGFR and E_Cd_/C_cr_ was seen in women and men with a low-Cd burden (E_Cd_/C_cr_ ≤ 0.014 µg/L filtrate in women and ≤0.012 µg/L filtrate in men) (Figure 1c,d). However, such eGFR/E_Cd_/C_cr_ relationship was weak and became statistically insignificant when covariates were adjusted. In comparison, eGFR showed an inverse association with E_Cd_/C_cr_ when E_Cd_/C_cr_ values rose to levels > 0.014 µg/L filtrate in women and >0.012 µg/L filtrate in men (Figure 1e,f and Table 3). In reflecting a dose–response relationship, eGFR was more closely associated with E_Cd_/C_cr_ in the high burden group (β = −0.349) compared to the medium Cd burden group (β = −0.284). An association of eGFR and E_Cd_/C_cr_ was statistically insignificant in the low Cd burden group.

Cd burden, diabetes and age were three of seven variables that affected the POR for low eGFR (Table 4). In those with medium, high Cd burden and diabetes, the POR for low eGFR rose by 8.3-fold, 3.6-fold and 3.0-fold, respectively. Per every one year rise in age, there was an 11.8% (95% CI, 6.2, 17.6) increase in the POR for low eGFR.

In summary, no effect of Cd on eGFR was found in those with E_Cd_/C_cr_ values < 0.012 µg/L filtrate. This E_Cd_/C_cr_ value of 0.012 µg/L filtrate is in ranges with the no-observed-adverse-effect level (NOAEL) equivalent of E_Cd_/C_cr_ at 0.010–0.024 µg/L filtrate, obtained by the benchmark dose method [44].

### 4.2. Effects of Cadmium on Prevalence of Albuminuria

Most albumin (~80%) in the glomerular ultrafiltrate is reabsorbed in the S1 sub-segment of the proximal tubule, where the receptor-mediated endocytosis involving the megalin/cubillin system is located [45,46,47]. Reabsorption of albumin also occurs in the distal tubule and collecting duct, where the process is mediated by the NGAL/lipocalin-2 receptor system [48,49,50]. Experimental studies show that albuminuria resulted from Cd selectively disabled albumin endocytosis which is mediated by the cubilin/megalin receptor system [51] and that Cd may diminish expression of megalin and ClC5 channels [52]. However, data in Table 1 indicate that albuminuria was more prevalent in the low eGFR compared to normal eGFR groups, thereby suggesting that low eGFR could be a risk factor for albuminuria. We examined this phenomenon further, as described below.

In multiple regressions (Table 5), both eGFR and E_Cd_/C_cr_ appeared to have an effect on E_alb_/C_cr_. In logistic regression (Table 6), the POR for albuminuria rose by 4.3% (95% CI: 2.6, 6.1) for every 1 mL/min/1.73 m^2^ loss of eGFR and it was increased by 2.4-fold to 2.7-fold in those who had [Cd]_b_ ≥ 0.83 µg/L. A similar effect size of Cd on eGFR was seen in women and men, although the causes of albuminuria differed (Table 6). Additional evidence that Cd may affect eGFR and albumin reuptake simultaneously comes from a covariance analysis, where 9.3% to 13.2% of E_alb_/C_cr_ variation could be attributable to eGFR levels in those with [Cd]_b_ > 0.83 µg/L (Figure 3d,f). eGFR levels did not explain the variability of E_alb_/C_cr_ in the low [Cd]_b_ tertile group (*F* = 0.674, η^2^ 0.010, *p* = 0.511) (Figure 3b).

The results described above could be interpreted to suggest that albuminuria could be a consequence of the Cd-induced destruction of nephrons, together with a direct effect on the tubular reabsorption of albumin, possibly through the cubilin/megalin receptor system of endocytosis. These Cd effects were not found in those who had [Cd]_b_ < 0.83 µg/L.

### 4.3. A Independent Effect of Smoking on Albuminuria

Smoking forms a significant Cd exposure source as cigarette smoke contains Cd in volatile metallic and oxide (CdO) forms, which have transmission rates 5 to 10 times higher than those that enter through the gut [53,54,55]. Adverse kidney outcomes associated with smoking have been noted in the general populations [56,57]. In the present study, evidence for an independent effect of smoking has emerged from analysis of covariance of albumin excretion data from those with normal kidney function. In Figure 2d, the mean E_alb_/C_cr_ was higher in smokers compared to non-smokers of the same age, BMI and overall Cd burden [(E_Cd_/C_cr_) × 100 ≥ 1 µg/L filtrate]. This finding suggested that a possible Cd independent effect of smoking on albuminuria. Other constitutes of cigarette smoke are likely to have a role. In a Dutch prospective study involving 231 diabetic patients, Cd and active smoking were both found to be associated with progressive eGFR decline [19].

Of further note, the opposite effect was observed in the low Cd burden group [(E_Cd_/C_cr_) × 100 < 1 µg/L filtrate] (Figure 2b), where the covariate adjusted mean E_alb_/C_cr_ was lower in smokers than nonsmokers of the same low Cd burden. Cd-induced albuminuria may be mitigated in smokers by substances that accompany Cd. Tin, for example, has blood pressure-lowering effects [58,59]. However, in our study, there was a high degree of statistical uncertainty due to a small number of smokers (*n* = 14) who had a low Cd burden (Figure 2b); hence, a further study is required to fully explore this.

An effect of diabetes on albumin excretion was also demonstratable in the present study; this effect was evident only in the low Cd burden group (Figure 2b). It is possible that we have underestimated the amount of albumin in urine samples from subjects with diabetes and high Cd burden (Figure 2d). The immunoturbidimetric method used has been reported to be unable to detect albumin molecules with altered conformation, which may be produced in people with CKD and diabetes [34].

### 4.4. Threshold-Based Exposure Guidelines

For most people, exposure to Cd is inevitable because the metal is present in nearly all food types. In Japan, for example, rice and its products, green vegetables and cereals and seeds plus potatoes constituted, respectively, to 38%, 17% and 11% of total dietary Cd exposure [60]. In response, health guidance such as a tolerable intake level of Cd and a reference dose (RfD) were determined [61,62,63,64]. The Joint FAO/WHO Expert Committee on Food Additives and Contaminants (JECFA) suggested a tolerable monthly intake (TMI) of Cd to be 25 μg per kg body weight per month, equivalent to 0.83 μg per kg body weight per day (58 µg/day for a 70 kg person), and E_Cd_/E_cr_ of 5.24 μg/g creatinine was adopted as a nephrotoxicity threshold value [63]. Both figures were based on a risk assessment model based solely on a β_2_-microglobulin excretion rate (E_β2M_/E_cr_) ≥ 300 μg/g creatinine, termed tubular proteinuria, as a toxic endpoint.

The European Food Safety Authority (EFSA) used also the β_2_M endpoint but with the inclusion of an uncertainty factor (safety margin). The EFSA suggested E_Cd_/E_cr_ of 1 μg/g creatinine to be the toxicity threshold and 0.36 μg/kg body weight per day (25 µg/day for a 70 kg person) as the RfD [64]. The higher JECFA guidelines are adopted by most countries. These health guideline values exceed the nephrotoxicity threshold identified in the present study. In theory, exposure guidelines that provide sufficient health protection should only be determined from the most sensitive endpoint with consideration given to subpopulations with increased susceptibility [54] and the most recent scientific knowledge should be considered.

Of concern, eGFR decline due to Cd nephropathy has increasingly been observed in both children and adult populations. Lower eGFR values were found to be associated with higher Cd excretion rates in studies from Guatemala [65] and Myanmar [66]. In a prospective cohort study of Bangladeshi preschool children, an inverse relationship between E_Cd_ and kidney volume was seen in children at 5 years of age. This was in addition to a decrease in eGFR [67]. E_Cd_ was inversely associated with eGFR, especially in girls. In another prospective cohort study, the reported mean for Cd intake among Mexican children was 4.4 µg/d at the baseline and rose to 8.1 µg/d after nine years, when such Cd intake levels showed a marginally inverse association with eGFR [68].

## 5. Conclusions

Based on a dose–response relationship analysis of data from 482 non-occupationally exposed persons with a 1250-fold difference in Cd burden and a 667-fold difference in levels of blood Cd, environmental exposure to Cd was confirmed to be closely associated with a declining GFR and albuminuria. An independent effect of smoking on albuminuria has also been observed in smokers who had normal kidney function. The current World Health Organization Cd exposure limit of 5.24 µg/g creatinine (E_Cd_/E_cr_), which is solely based on an increment of excretion of β_2_-microglobulin above 300 µg/g creatinine, underestimates the level at which Cd induces kidney damage. Our results show that when a declining GFR is considered along with albuminuria, the no-observed-adverse-effect level (NOAEL) equivalent is 0.01–0.02 µg/g creatinine. Now is the time to acknowledge there is no safe level of Cd exposure.

## Figures and Tables

**Figure 1 toxics-11-00755-f001:**
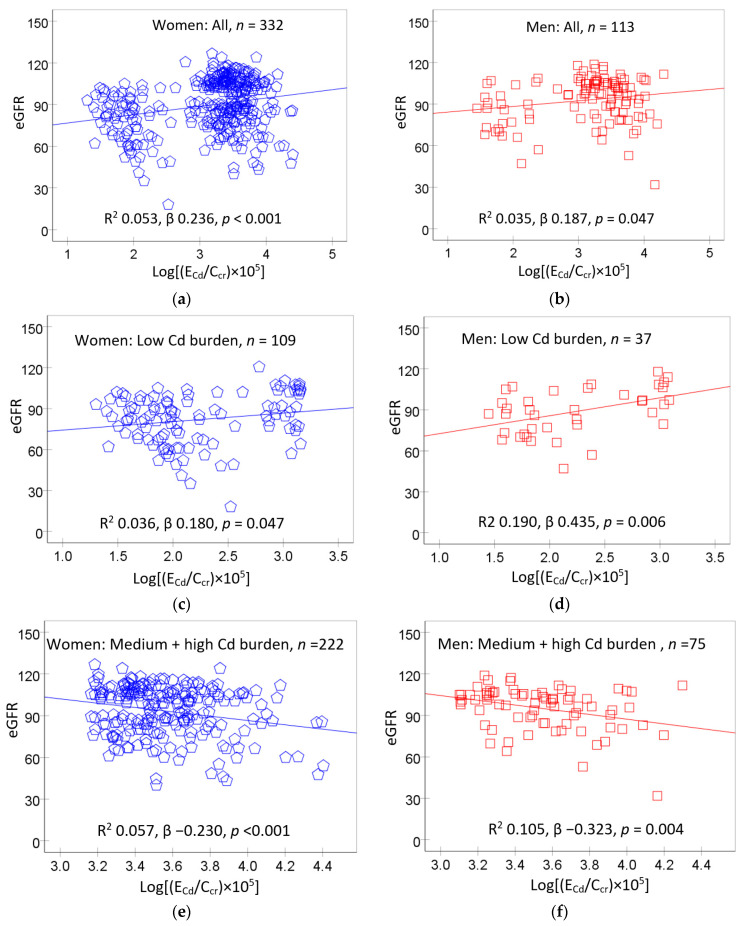
The relationship between eGFR and cadmium excretion rate. Scatterplots relate eGFR to log[(E_Cd_/C_cr_) × 10^5^) in women (**a**) and men (**b**), women (**c**) and men (**d**) with low Cd burden and women (**e**) and men (**f**) with medium plus high Cd burden. Coefficients of determination (R^2^) and *p*-values are provided for all scatterplots together with numbers of participants in all subgroups. The cut-off values of (E_Cd_/C_cr_) × 100 for low, medium and high Cd burden in women were ≤1.44, 1.45–3.26, >3.26 µg/L filtrate, respectively. Corresponding cut-off values of (E_Cd_/C_cr_) × 100 in men were ≤1.25, 1.26–3.25, >3.25 µg/L filtrate.

**Figure 2 toxics-11-00755-f002:**
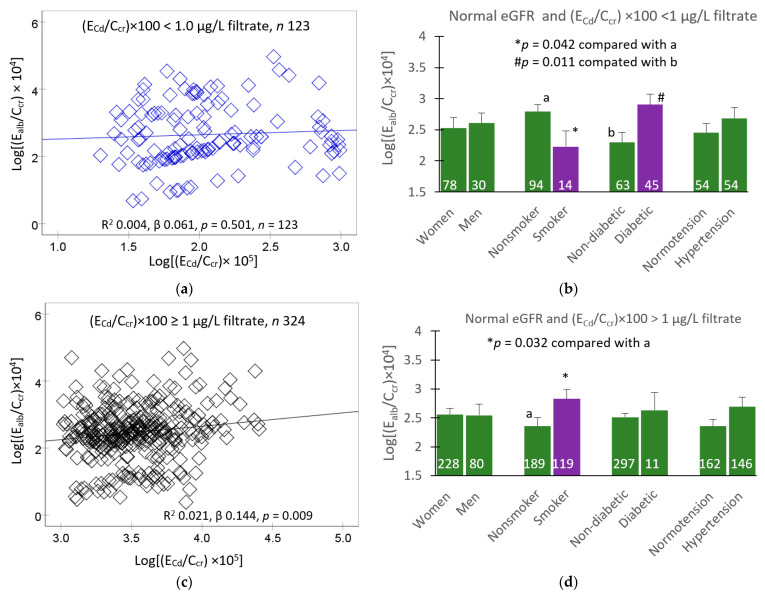
Albumin excretion rates among participants grouped by kidney function, cadmium burden and other risk factors. Scatterplots relate log[(E_alb_/C_cr_) × 10^4^] to log[E_Cd_/C_cr_) × 10^5^] in participants with (E_Cd_/C_cr_) × 100 < 1 and ≥ 1 µg/L filtrate (**a**,**c**). Coefficients of determination (R^2^) and *p*-values and numbers of participants are provided. Bar graphs depict mean log[(E_alb_/C_cr_) × 104] in participants with normal eGFR and (E_Cd_/C_cr_) × 100 < 1 and ≥ 1 µg/L filtrate (**b**,**d**). Normal eGFR was defined as eGFR > 60 mL/min/1.73 m^2^. Numbers of subjects are provided for all subgroups. All means were obtained via univariate covariance analysis with adjustment for covariates (age and BMI) and (smoking × hypertension × diabetes) interactions.

**Figure 3 toxics-11-00755-f003:**
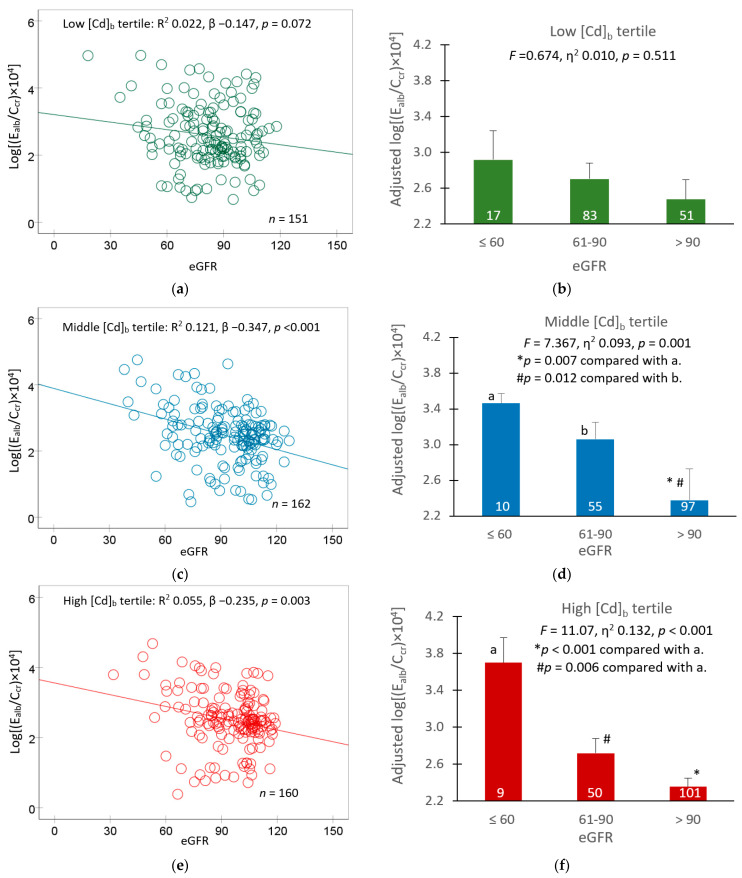
Albumin excretion rates in participants grouped by blood cadmium and eGFR levels. Scatterplots relate log[(E_alb_/C_cr_) × 10^4^] to eGFR in participants with low (**a**), middle (**c**) and high [Cd]_b_ tertiles (**e**). Coefficients of determination (R^2^), *p*-values and numbers of subjects are provided for all scatterplots. Bar graphs depict mean log[(E_alb_/C_cr_) ×10^4^] in participants of low (**b**), middle (**d**) and high [Cd]_b_ tertiles (**f**) who had eGFR ≤ 60, 61–90 and > 90 mL/min/1.73 m^2^. All means were obtained via univariate covariance analysis with adjustment for covariates and interactions. The cut-off values of [Cd]_b_ for low, middle and high tertiles were < 0.82, 0.83–2.63 and ≥ 2.64 µg/L, respectively. Arithmetic means (SD) of [Cd]_b_ in low, middle and high tertiles were 0.31 (0.29), 1.86 (1.98) and 5.58 (3.28), respectively. For all tests, *p*-values ≤ 0.05 indicate statistically significant levels.

**Table 1 toxics-11-00755-t001:** Characteristics of participants according to eGFR levels and gender.

Parameters	All *n* = 482	Normal eGFR ^a^		Low eGFR	
		Women *n* = 329	Men *n* = 114	Women *n* = 35	Men *n* = 4
Age (years)	51.8 ± 9.2	51.2 ± 8.7	49.8 ± 7.5	63.1 ± 11.3	56.0 ± 6.4
BMI (kg/m^2^)	24.8 ± 4.0	25.3 ± 4.0	23.5 ± 3.7 ***	24.0 ± 3.6	26.7 ± 5.2
eGFR ^b^, mL/min/1.73 m^2^	90 ± 19	93 ± 15	94 ± 14	51 ± 9	47 ± 11
Smoking (%)	29.7	17.6	68.4 ***	11.4	75.0 ##
Diabetes (%)	18.3	16.7	13.2	45.7	50.0
Hypertension (%)	48.3	49.8	40.4	54.3	100
Systolic blood pressure	129 ± 17	129 ± 16	127 ± 113	138 ± 18	147 ± 5
Diastolic blood pressure	81 ± 10	81 ± 10	81 ± 11	79 ± 8	86 ± 4
Serum creatinine, mg/dL	0.82 ± 0.22	0.74 ± 0.13	0.92 ± 0.14 ***	1.19 ± 0.31	1.68 ± 0.43 ##
Urine creatinine, mg/dL	113 ± 73	110 ± 74	134 ± 72 ***	72 ± 33	73 ± 24
Urine albumin, mg/L	21 ± 57	13 ± 31	29 ± 77	60 ± 118	86 ± 106
Blood Cd, µg/L	2.60 ± 3.13	2.43 ± 2.96	3.13 ± 3.32 *	2.19 ± 3.67	5.03 ± 5.03
Urine Cd, µg/L	4.21 ± 5.66	4.44 ± 6.15	3.84 ± 4.04	3.31 ± 5.71	2.26 ± 2.80
Normalized to E_cr_ (E_x_/E_cr_) ^c^					
ACR, mg/g creatinine	24 ± 71	15 ± 41	27 ± 70	93 ± 181	112 ± 139
Albuminuria (%) ^d^	15.0	11.4	17.0	34.4	100 #
E_Cd_/E_cr_, µg/g creatinine	4.05 ± 4.43	4.27 ± 4.46	3.32 ± 3.71 *	4.63 ± 6.31	2.62 ± 3.04
Normalized to C_cr_, (E_x_/C_cr_) ^e^					
(E_alb_/C_cr_) × 100, mg/L	24 ± 85	12 ± 32	26 ± 67	125 ± 252	174 ± 205
Abnormal E_alb_/C_cr_ (%) ^f^	16.9	12.9	17.0	46.9	100 #
(E_Cd_/C_cr_) × 100, µg/L	3.20 ± 3.73	3.05 ± 3.25	3.05 ± 3.41	5.14 ± 7.44	5.20 ± 6.80

*n*, number of subjects; eGFR, estimated glomerular filtration rate; BMI, body mass index; ACR, albumin-to-creatinine ratio; alb, albumin. ^a^ The cutoff eGFR of 60 mL/min/1.73 m^2^ was used to define departure from normalcy. ^b^ eGFR was determined with equations of the Chronic Kidney Disease Epidemiology Collaboration (CKD−EPI) [35]. ^c^ E_x_/E_cr_ = [x]_u_/[cr]_u_, where x = alb or Cd. ^d^ Albuminuria is defined as ACR ≥ 20 mg/g for men and ≥30 mg/g for women. ^e^ E_x_/C_cr_ = [x]_u_[cr]_p_/[cr]_u_, where x = alb or Cd [23]. ^f^ Abnormal E_alb_/C_cr_ was defined as [(E_alb_/C_cr_) × 100] ≥ 20 mg/L of filtrate. Data for all continuous variables are arithmetic mean ± standard deviation (SD). For all tests, *p* ≤ 0.05 identifies statistical significance, determined with the Pearson Chi-Square test for differences between percentages and the Mann–Whitney U test for differences of means. For the normal eGFR group, *** *p* < 0.001; * *p* = 0.016–0.021. For the low eGFR group, ## *p* = 0.002; # *p* = 0.012–0.045.

**Table 2 toxics-11-00755-t002:** Predictors of eGFR in women and men.

Independent Variables/Factors	eGFR, mL/min/1.73 m^2^					
	All, *n* = 444		Women, *n* = 332		Men, *n* = 113	
	β	*p*	β	*p*	β	*p*
Age, years	−0.503	<0.001	−0.511	<0.001	−0.472	<0.001
BMI, kg/m^2^	−0.066	0.129	−0.050	0.310	−0.134	0.149
Log[(E_Cd_/C_cr_) × 10^5^], µg/L filtrate	−0.121	0.022	−0.126	0.043	−0.097	0.367
Systolic pressure, mmHg	−0.087	0.134	−0.067	0.320	−0.147	0.230
Diastolic pressure, mmHg	−0.011	0.836	−0.020	0.745	0.022	0.856
Gender	−0.006	0.898	−	−	−	−
Smoking	0.033	0.492	0.031	0.533	0.034	0.710
Diabetes	−0.101	0.038	−0.119	0.036	−0.033	0.741
Adjusted R^2^	0.278	<0.001	0.279	<0.001	0.216	<0.001

eGFR, estimated glomerular filtration rate; β, standardized regression coefficient; adjusted R^2^, coefficient of determination. Coding: male, 1, female, 2; non-smoker, 1, smoker, 2; non-diabetic, 1, diabetic, 2. β indicates strength of association of eGFR with eight independent variables (first column). Adjusted R^2^ indicates the proportion of eGFR variance explained by all independent variables. For all tests, *p*-values ≤ 0.05 indicate a statistically significant effect of individual independent variables on eGFR.

**Table 3 toxics-11-00755-t003:** Comparing strength of association of eGFR with cadmium excretion rate across cadmium burden groups.

Independent Variables/Factors	eGFR, mL/min/1.73 m^2^					
	Low Cd Burden		Medium Burden		High Burden	
	β	*p*	β	*p*	β	*p*
Age, years	−0.455	<0.001	−0.505	<0.001	−0.410	<0.001
BMI, kg/m^2^	−0.022	0.777	−0.136	0.064	−0.094	0.221
Log[(E_Cd_/C_cr_) × 10^5^], µg/L filtrate	0.007	0.934	−0.230	0.002	−0.349	<0.001
Systolic pressure, mmHg	−0.038	0.722	−0.284	0.004	−0.050	0.611
Diastolic pressure, mmHg	0.087	0.372	0.113	0.226	−0.170	0.078
Gender	−0.024	0.798	−0.064	0.434	0.106	0.194
Smoking	0.094	0.324	−0.081	0.301	0.132	0.108
Diabetes	−0.128	0.104	0.089	0.260	0.022	0.775
Adjusted R^2^	0.248	<0.001	0.321	<0.001	0.255	<0.001

eGFR, estimated glomerular filtration rate; β, standardized regression coefficient; adjusted R^2^, coefficient of determination. Coding: male, 1, female, 2; non-smoker, 1, smoker, 2; non-diabetic, 1, diabetic, 2. β indicates strength of association of eGFR with eight independent variables (first column). Adjusted R^2^ indicates the proportion of eGFR variance explained by all independent variables. The cut-off values of (E_Cd_/C_cr_) × 100 for the low, medium and high Cd burdens in women were ≤1.44, 1.45–3.26, >3.26 µg/L filtrate, respectively. Corresponding cut-off values of (E_Cd_/C_cr_) × 100 in men were ≤1.25, 1.26–3.25, >3.25 µg/L filtrate. The numbers of subjects in low, moderate and high Cd burdens were 147, 148 and 147, respectively. For all tests, *p*-values ≤ 0.05 indicate a statistically significant effect of individual independent variables on eGFR.

**Table 4 toxics-11-00755-t004:** Prevalence odds ratios for low eGFR in relation to cadmium body burden and other independent variables.

Independent Variables/Factors	Low eGFR					
	All, *n* = 446		Women, *n* = 332		Men, *n* = 114	
	POR (95% CI)	*p*	POR (95% CI)	*p*	POR (95% CI)	*p*
Age, years	1.118 (1.062, 1.176)	<0.001	1.114 (1.057, 1.175)	<0.001	1.291 (0.935, 1.783)	0.120
BMI, kg/m^2^	1.002 (0.908, 1.106)	0.967	1.029 (0.923, 1.147)	0.604	1.351 (0.876, 2.084)	0.174
Gender	0.482 (0.133, 1.742)	0.265	−	−	−	−
Smoking	1.388 (0.433, 4.450)	0.582	1.123 (0.300, 4.206)	0.863	0.495 (0.018, 13.92)	0.679
Diabetes	3.042 (1.126, 8.213)	0.028	2.709 (0.932, 7.878)	0.067	3.713 (0.110, 125.6)	0.465
Hypertension	1.175 (0.516, 2.679)	0.701	1.228 (0.505, 2.990)	0.651	2.478 (1.983, 3.098)	0.030 ^a^
Cd body burden						
Low	Referent		Referent		Referent	
Medium	8.265 (1.711, 39.92)	0.009	7.204 (1.438, 36.10)	0.016	n/a	n/a
High	3.643 (1.150, 11.54)	0.028	3.218 (0.934, 11.09)	0.064	n/a	n/a

POR, prevalence odds ratio; CI, confidence interval; BMI, body mass index; eGFR, estimated glomerular filtration rate. Coding, male 1, female 2; non-smoker 1, smoker 2; non-diabetic 1, diabetic 2; code 1 is referent. Data were generated from logistic regression relating POR for low eGFR to seven independent variables (first column). ^a^ Effect of hypertension was assessed by Fisher’s exact test using data from 118 men. The cut-off values of (E_Cd_/C_cr_) × 100 for the low, medium and high Cd burdens in women were ≤1.44, 1.45–3.26, >3.26 µg/L filtrate, respectively. Corresponding cut-off values of (ECd/Ccr) × 100 in men were ≤1.25, 1.26–3.25, >3.25 µg/L filtrate. For all tests, *p*-values ≤ 0.05 indicate a statistically significant effect of individual independent variables to the POR for low eGFR.

**Table 5 toxics-11-00755-t005:** Comparing strength of association of albumin excretion rate with cadmium excretion rate across cadmium burden groups.

Independent Variables/Factors	Log [(E_alb_/C_cr_) × 10^4^], µg/L Filtrate					
	Low Cd Burden		Medium Burden		High Burden	
	β	*p*	β	*p*	β	*p*
Age, years	0.049	0.609	−0.019	0.846	−0.082	0.381
BMI, kg/m^2^	−0.007	0.932	0.248	0.004	−0.021	0.803
Log[(E_Cd_/C_cr_) × 10^5^], µg/L filtrate	0.177	0.050	0.144	0.098	0.173	0.044
eGFR, mL/min/1.73 m^2^	−0.110	0.216	−0.147	0.130	−0.214	0.021
Gender	−0.150	0.122	−0.111	0.233	0.139	0.120
Smoking	−0.183	0.064	0.024	0.788	0.129	0.153
Diabetes	0.263	0.001	0.093	0.280	0.115	0.176
Hypertension	0.278	<0.001	0.121	0.148	0.104	0.204
Adjusted R^2^	0.174	<0.001	0.103	0.003	0.082	0.009

eGFR, estimated glomerular filtration rate; β, standardized regression coefficient; adjusted R^2^, coefficient of determination. Coding: male, 1, female, 2; non-smoker, 1, smoker, 2; non-diabetic, 1, diabetic, 2. β indicates strength of association of eGFR with eight independent variables (first column). Adjusted R^2^ indicates the proportion of eGFR variance explained by all independent variables. The cut-off values of (E_Cd_/C_cr_) × 100 for the low, medium and high Cd burdens in women were ≤1.44, 1.45–3.26, >3.26 µg/L filtrate, respectively. Corresponding cut-off values of (E_Cd_/C_cr_) × 100 in men were ≤1.25, 1.26–3.25, >3.25 µg/L filtrate. The numbers of subjects in low, medium and high Cd burdens were 147, 148 and 149, respectively. For all tests, *p*-values ≤ 0.05 indicate a statistically significant effect of individual independent variables on eGFR.

**Table 6 toxics-11-00755-t006:** Comparing strength of association of albumin excretion rate with cadmium excretion rate and other variables in participants grouped by eGFR.

Independent Variables/Factors	Log [(E_alb_/C_cr_) × 10^4^], µg/L Filtrate					
	All		Normal eGFR		Low eGFR	
	β	*p*	β	*p*	β	*p*
Age, years	−0.023	0.710	−0.016	0.803	0.163	0.373
BMI, kg/m^2^	0.053	0.265	0.033	0.522	0.363	0.040
Log[(E_Cd_/C_cr_) × 10^5^], µg/L filtrate	0.104	0.078	0.041	0.511	0.637	0.008
eGFR, mL/min/1.73 m^2^	−0.188	<0.001	−0.072	0.193	−0.099	0.505
Gender	−0.010	0.851	0.002	0.974	−0.084	0.590
Smoking	0.026	0.624	0.010	0.863	0.014	0.934
Diabetes	0.219	<0.001	0.173	0.002	0.561	0.006
Hypertension	0.167	<0.001	0.150	0.002	0.246	0.130
Adjusted R^2^	0.115	<0.001	0.047	0.002	0.522	0.001

eGFR, estimated glomerular filtration rate; β, standardized regression coefficient; adjusted R^2^, coefficient of determination. Coding: male, 1, female, 2; non-smoker, 1, smoker, 2; non-diabetic, 1, diabetic, 2. β indicates strength of association of log[(E_alb_/C_cr_) × 10^4^] with eight independent variables (first column). Adjusted R^2^ indicates the proportion of log[(E_alb_/C_cr_) × 10^4^] variance explained by all independent variables. Normal and low eGFR groups were defined as eGFR > 60 and ≤60 mL/min/1.73 m^2^, respectively. Respective numbers of subjects with normal and low eGFR were 416 and 31. For all tests, *p*-values ≤ 0.05 indicate a statistically significant effect of individual independent variables on albumin excretion rate.

**Table 7 toxics-11-00755-t007:** Prevalence odds ratios for albuminuria in relation to blood Cd levels and other independent variables.

Independent Variables/Factors	Albuminuria, (E_alb_/C_cr_) ×100 ≥ 20 mg/L Filtrate					
	All, *n* = 473		Women, *n* = 357		Men, *n* = 116	
	POR (95% CI)	*p*	POR (95% CI)	*p*	POR (95% CI)	*p*
Age, years	0.990 (0.955, 1.027)	0.609	1.007 (0.965, 1.051)	0.733	0.988 (0.904, 1.079)	0.781
BMI, kg/m^2^	1.028 (0.957, 1.104)	0.448	1.033 (0.950, 1.123)	0.450	1.011 (0.877, 1.166)	0.881
eGFR, mL/min/1.73 m^2^	1.043 (1.026, 1.061)	<0.001	1.045 (1.025, 1.065)	<0.001	1.045 (1.008, 1.084)	0.017
Gender	0.559 (0.273, 1.146)	0.112	−	−	−	−
Smoking	1.037 (0.494, 2.177)	0.924	1.258 (0.496, 3.192)	0.629	0.809 (0.254, 2.572)	0.719
Diabetes	6.021 (2.813, 12.89)	<0.001	5.996 (2.446, 14.69)	<0.001	8.324 (1.642, 42.21)	0.011
Hypertension	2.053 (1.167, 3.609)	0.013	2.785 (1.397, 5.552)	0.004	1.133 (0.371, 3.463)	0.827
Tertile of [Cd]_b_, µg/L						
Low: <0.82	Referent		Referent		Referent	
Middle: 0.83–2.63	2.360 (1.097, 5.076)	0.028	3.402 (1.324, 8.745)	0.011	0.925 (0.237, 3.604)	0.911
High: ≥2.64	2.740 (1.174, 6.394)	0.020	3.783 (1.369, 10.46)	0.010	1.425 (0.263, 7.732)	0.681

POR, prevalence odds ratio; CI, confidence interval; BMI, body mass index; eGFR, estimated glomerular filtration rate; [Cd]_b_, blood Cd concentration. Coding, male 1, female 2; non-smoker 1, smoker 2; non-diabetic 1, diabetic 2; code 1 is referent. Data were generated from logistic regression relating POR for albuminuria to seven independent variables (first column). Arithmetic means (SD) of blood Cd in low, middle and high tertiles were 0.31 (0.29), 1.86 (1.98), 5.58 (3.28), respectively. Corresponding numbers of subjects were 157, 164 and 161, respectively. For all tests, *p*-values ≤ 0.05 indicate a statistically significant effect of individual independent variables to the POR for albuminuria.

## Data Availability

All data are contained within this article.

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
