# Peer review of "Estimation of the Cadmium Nephrotoxicity Threshold from Loss of Glomerular Filtration Rate and Albuminuria"

_toxics, 2023, doi:10.3390/toxics11090755_

Round 1
Reviewer 1 Report
I have admired and read the work of this group of collaborators going back many years to some of Dr. Satarug's early publications. This manuscript is very well written and the research plan well thought out. The data presented in this manuscript supports the authors' conclusion statement, "... there is no safe level of Cd exposure.". I hope that my comments will enable improvement in an already good manuscript.
1. Lines 118-119: "The limit of detection (LOD) of Cd, defined as 3 times the standard deviation of blank measurements": The authors should state how many blank measurements were used to calculate the LOD and what type of blanks were used.
2. Line 123: The authors should relate more information on the turbidometric method used for determination of urinary albumin.
3. Lines 132-135: I note that the cutoff level for assessment of chronic kidney disease is stage 3 (d eGFR ≤ 60 mL/min/1.73 m2). This is reasonable. I only note this for later comment.
4. Lines 147-148: I note that statistical comparisons were between different opposing groups. This is reasonable. I note this only for later comment.
5. I note in Table 2, Table 4, and lines 197 - 199, 415-416 that Age, ECd/Ccr, and diabetic condition are the principal factors observed to correlate with decreases in eGFR, albuminuria, etc., whereas smoking, which is a major source of cadmium exposure, is not. When smoking status is considered as the sole smoking related variable for statistical analysis, it is not surprising that the effect on the major impact of smoking as a source of cadmium exposure was not significant, whereas age was significant. In Table 3 of Paschal et al. (Exposure of the U.S. Population Aged 6 Years and Older to Cadmium: 1988–1994, Arch. Environ. Contam. Toxicol. 38, 377–383 (2000) DOI: 10.1007/s002449910050) showed that urinary cadmium excretion in a non-industrially exposed population increases with age (indicating long term chronic exposure) whether individuals smoke or not, which correlates with the age related decrease in eGFR and CKD reported in the manuscript submitted for consideration here. However, in Table 3, one also notes that among a population not otherwise environmentally exposed to high levels of cadmium pollution, the rate of increase in urinary cadmium concentration among smokers is much greater than among nonsmokers. I believe that the authors' choice of how to assess the impact of smoking is not optimally considered. There are two populations: Elevated environmental exposure including high rice paddy water cadmium concentrations and consequential high dietary cadmium concentrations (rice, etc.). In order to best represent the effect of smoking on urinary cadmium concentrations correlated with smoking, the groups should be further broken down. Urinary cadmium increases with age among low environmental exposure and high environmental individuals because of length of time for bioaccumulation. By the same reasoning, A young smoker has had less time to accumulate cadmium than a person who has smoked for 10 or 20 years. When smoking is considered as a single factor without regard to age or number of pack years smoked, the range of urinary cadmium concentrations will be large. When this is compounded by inclusion of low and high dietary and water exposures, it is not surprising that the statistics, which only analyze what the program parameters determine based on authors' decisions, show no significant correlation with smoking. Smoking is also a factor in prevalence of hypertension and diabetes, which were significantly correlated with urinary cadmium concentrations, eGFR decreases, albuminuria, etc.
The authors should therefore reexamine smoking in consideration of the following. A person who has smoked for a short time (age) but is not highly environmentally exposed (diet, water, etc.) may not have higher cadmium related pathology than a person who does not smoke but has a much higher environmental exposure. Therefore, if the authors divide smokers among short term exposure, perhaps under ager 30 versus age 30-40, versus over age 40, a significant difference may emerge. If the authors further analyze differences in urinary cadmium, eGFR, albuminuria among young smokers and older smokers in the low environmentally exposed area, and differences among young smokers and older smokers in the high environmentally exposed area, thus removing the variability in three variables (age, environmental exposure, smoking status to one variable, I believe the authors may have additional highly impactful results to report.
Author Response
Reviewer 1
I have admired and read the work of this group of collaborators going back many years to some of Dr. Satarug's early publications. This manuscript is very well written and the research plan well thought out. The data presented in this manuscript supports the authors' conclusion statement, "... there is no safe level of Cd exposure.". I hope that my comments will enable improvement in an already good manuscript.
RESPONSE: We thank the reviewer for insightful comments and guidance to further improve our manuscript. Accordingly, below we provide point-by-point responses to each comment. Changes to the manuscript text are in blue.
Comment 1. Lines 118-119: "The limit of detection (LOD) of Cd, defined as 3 times the standard deviation of blank measurements": The authors should state how many blank measurements were used to calculate the LOD and what type of blanks were used.
RESPONSE: We have inserted statements given below to describe how the LOD was calculated (lines 122-126).
“The limit of detection (LOD) for Cd in blood or urine, defined as 3 times the standard deviation of at least 10 blank sample measurements was 0.3 µg/L for [Cd]b and 0.1 µg/L for [Cd]u. The sample blanks, reference standards for urine and blood were included in the assay together with samples of urine or blood from subjects. Deionized water was used to zero an instrument.”
Comment 2. Line 123: The authors should relate more information on the turbidometric method used for determination of urinary albumin.
RESPONSE: We have provided below references for the turbidimetric method used for urinary albumin determination. The imprecision of this method of urinary albumin measurement is discussed in the Discussion (lines 492-497).
[33] Bargnoux, A.S.; Barrot, A.; Fesler, P.; Kuster, N.; Badiou, S.; Dupuy, A.M.; Ribstein, J.; Cristol, J.P. Evaluation of five immunoturbidimetric assays for urinary albumin quantification and their impact on albuminuria categorization. Clin. Biochem. 2014, 47, 250-253.
[34] Kok, M.B.; Tegelaers, F.P.; van Dam, B.; van Rijn, J.L.; van Pelt, J. Carbamylation of albumin is a cause for discrepancies between albumin assays. Clin. Chim. Acta 2014, 434, 6-10.
Comment 3. Lines 132-135: I note that the cutoff level for assessment of chronic kidney disease is stage 3 (eGFR ≤ 60 mL/min/1.73 m2). This is reasonable. I only note this for later comment.
Comment 4. Lines 147-148: I note that statistical comparisons were between different opposing groups. This is reasonable. I note this only for later comment.
Comment 5. I note in Table 2, Table 4, and lines 197 - 199, 415-416 that Age, ECd/Ccr, and diabetic condition are the principal factors observed to correlate with decreases in eGFR, albuminuria, etc., whereas smoking, which is a major source of cadmium exposure, is not. When smoking status is considered as the sole smoking related variable for statistical analysis, it is not surprising that the effect on the major impact of smoking as a source of cadmium exposure was not significant, whereas age was significant. In Table 3 of Paschal et al. (Exposure of the U.S. Population Aged 6 Years and Older to Cadmium: 1988–1994, Arch. Environ. Contam. Toxicol. 38, 377–383 (2000) DOI: 10.1007/s002449910050) showed that urinary cadmium excretion in a non-industrially exposed population increases with age (indicating long term chronic exposure) whether individuals smoke or not, which correlates with the age-related decrease in eGFR and CKD reported in the manuscript submitted for consideration here. However, in Table 3, one also notes that among a population not otherwise environmentally exposed to high levels of cadmium pollution, the rate of increase in urinary cadmium concentration among smokers is much greater than among nonsmokers. I believe that the authors' choice of how to assess the impact of smoking is not optimally considered. There are two populations: Elevated environmental exposure including high rice paddy water cadmium concentrations and consequential high dietary cadmium concentrations (rice, etc.). In order to best represent the effect of smoking on urinary cadmium concentrations correlated with smoking, the groups should be further broken down. Urinary cadmium increases with age among low environmental exposure and high environmental individuals because of length of time for bioaccumulation. By the same reasoning, A young smoker has had less time to accumulate cadmium than a person who has smoked for 10 or 20 years. When smoking is considered as a single factor without regard to age or number of pack years smoked, the range of urinary cadmium concentrations will be large. When this is compounded by inclusion of low and high dietary and water exposures, it is not surprising that the statistics, which only analyze what the program parameters determine based on authors' decisions, show no significant correlation with smoking. Smoking is also a factor in prevalence of hypertension and diabetes, which were significantly correlated with urinary cadmium concentrations, eGFR decreases, albuminuria, etc.
The authors should therefore reexamine smoking in consideration of the following. A person who has smoked for a short time (age) but is not highly environmentally exposed (diet, water, etc.) may not have higher cadmium related pathology than a person who does not smoke but has a much higher environmental exposure. Therefore, if the authors divide smokers among short term exposure, perhaps under ager 30 versus age 30-40, versus over age 40, a significant difference may emerge. If the authors further analyze differences in urinary cadmium, eGFR, albuminuria among young smokers and older smokers in the low environmentally exposed area, and differences among young smokers and older smokers in the high environmentally exposed area, thus removing the variability in three variables (age, environmental exposure, smoking status to one variable, I believe the authors may have additional highly impactful results to report.
RESPONSE TO COMMENTS 3, 4 and 5
- We were grateful to the guidance the reviewer has provided. We have now carefully studied and cited the reference Paschal et al. (Exposure of the U.S. Population Aged 6 Years and Older to Cadmium: 1988–1994, Arch. Environ. Contam. Toxicol. 38, 377–383 (2000).
- We agree with the reviewer that demonstration of the impact of smoking in the present study is necessary, given that cigarette smoke is known to contain Cd in volatile metallic and oxide (CdO) forms which have transmission rates 5- to 10-times higher than those enter through the gut.
- We agree with the reviewer that delineation of the effects of smoking would require subgrouping of cohort participants according to age, smoking status and exposure through diet or diet plus smoking. However, due to a small number of participants from the low exposure locality (174 of total 482) and a low smoking prevalence of 9.8%, compared to 40.9% among those lived in a Cd contaminated area, the small subgroups (n < 25) had insufficient statistical power for covariance analysis (ANCOVA). We thus undertook an alternative approach as described below.
- A new Table 6 has been included which shows the effect of kidney function (GFR) on the albumin excretion rate (lines 321-330). After controlling for this GFR effect through restricting analysis to those with normal eGFR, an independent effect of smoking on albumin excretion became apparent in those with (ECd/Ccr) ×100 >1 µg/L filtrate. This result is reported in a new Figure 2.
- A new paragraph, 4.3. An Independent Effect of Smoking on Albuminuria, has been inserted in the discussion (lines 473-497) to report the effect of smoking on the albumin excretion rate, which was obscure when subjects with normal kidney function and those with CKD were analyzed together. This new paragraph is quoted below.
- This additional finding on the effects of smoking has been included in the conclusion. We thank the reviewer for suggesting we reexamine the effects of smoking.
- References to smoking as a source of Cd and the effects of smoking on CKD (eGFR and albuminuria criterion) have been added.
4.3. The Independent Effect of Smoking on Albuminuria
Smoking forms a significant Cd source as cigarette smoke contains Cd in volatile metallic and oxide (CdO) forms, which have transmission rates 5 to 10 times higher than those that enter through the gut [53-55]. Adverse kidney outcomes associated with smoking have been noted in the general populations [56,57]. In the present study, evidence for an independent effect of smoking has emerged from a covariance analysis of on albumin excretion data from those with normal kidney function. In Figure 2d, the mean Ealb/Ccr was higher in smokers, compared to non-smokers of the same age, BMI and overall Cd burden [(ECd/Ccr) ×100 ≥ 1 µg/L filtrate]. This finding suggested that the effect of smoking on albuminuria which may be mediated by other constitutes of cigarette smoke. In a Dutch prospective study involving 231 diabetic patients, Cd and active smoking both were found to be associated with progressive eGFR decline [19].
Of further note, the opposite effect was observed in the low Cd burden group [(ECd/Ccr) ×100 < 1 µg/L filtrate] (Figure 2b), where the covariate-adjusted mean Ealb/Ccr was lower in smokers than nonsmokers of the same low Cd burden. Cd-induced albuminuria may be mitigated in smokers by substances that accompany Cd. Tin, for example, has blood pressure-lowering effects [58, 59]. However, a high degree of statistical uncertainty due to a small number of smokers (n = 14) who had low Cd burden (Figure 2b), a further study is required.
An effect of diabetes on albumin excretion was also demonstratable in the present study, but this effect was evident only in the low Cd burden group (Figure 2b). It was likely that we underestimated the amount of albumin in urine samples from subjects with diabetes and high Cd burden (Figure 2d). The immunoturbidimetric method used failed to detect albumin molecules with altered conformation that could have been produced in people with CKD and diabetes [34].
[53] Pappas, R.S.; Fresquez, M.R.; Watson, C.H. Cigarette smoke cadmium breakthrough from traditional filters: Implications for exposure. J. Anal. Toxicol. 2015, 39, 45–51.
[54] Paschal, D.C.; Burt, V.; Caudill, S.P.; Gunter, E.W.; Pirkle, J.L.; Sampson, E.J.; Miller, D.T.; Jackson, R.J. Exposure of the U.S. population aged 6 years and older to cadmium: 1988-1994. Arch. Environ. Contam. Toxicol. 2000, 38, 377-383.
[55] Mortensen, M.E.; Wong, L.Y.; Osterloh, J.D. Smoking status and urine cadmium above levels associated with subclinical renal effects in U.S. adults without chronic kidney disease. Int. J. Hyg. Environ. Health 2011, 214, 305-310.
[56] Kelly, J.T.; Su, G.; Zhang, L.; Qin, X.; Marshall, S.; González-Ortiz, A.; Clase, C.M.; Campbell, K.L.; Xu, H.; Carrero, J.J. Modifiable lifestyle factors for primary prevention of CKD: A systematic review and meta-analysis. J. Am. Soc. Nephrol. 2021, 32, 239–253.
[57] Jin, A.; Koh, W.P.; Chow, K.Y.; Yuan, J.M.; Jafar, T.H. Smoking and risk of kidney failure in the Singapore Chinese health study. PLoS ONE 2013, 8, e62962.
[58] Sacerdoti, D.; Escalante, B.; Abraham, N.G.; McGiff, J.C.; Levere, R.D.; Schwartzman, M.L. Treatment with tin prevents the development of hypertension in spontaneously hypertensive rats. Science 1989, 243, 388-390.
[59] Escalante, B.; Sacerdoti, D.; Davidian, M.M.; Laniado-Schwartzman, M.; McGiff, J.C. Chronic treatment with tin normalizes blood pressure in spontaneously hypertensive rats. Hypertens. 1991, 17, 776-779.

Reviewer 2 Report
Authors submitted a manuscript titled "Estimation of the Cadmium Nephrotoxicity Threshold from 2 Loss of GFR and Albuminuria" for publication in Toxics journal. The objective was to quantify changes in estimated GFR (eGFR) and albumin excretion (Ealb) according to levels 19 of blood Cd ([Cd]b) and excretion of Cd (ECd) after adjustment for confounders among 482 residents of Cd-polluted and non-polluted regions of Thailand. ECd and Ealb were normalized to creatinine clearance (Ccr) as ECd/Ccr and Ealb/Ccr. Obtained results indicate that ECd/Ccr of 1.44 µg/L filtrate (0.01−0.02 µg/g creatinine) could determine a Cd threshold level from which protective exposure guidelines should be formulated. The idea is good. It should be tested through further investigation in another population even if the power of study is relatively strong. The authors were focused on the dose-response relationship, a critical step in the hazard assessment. Moreover, univariate analysis was used to quantify the independent effects of ECd/Ccr, eGFR, diabetes, and hypertension on Ealb/Ccr as well as blood Cd and eGFR as predictors of albuminuria. Based on the results, the authors are pointing to the observation that there are no safe levels of Cd exposure.
I suggest after proof reading the publication of a proposed manuscript.
Author Response
Reviewer 2
Authors submitted a manuscript titled "Estimation of the Cadmium Nephrotoxicity Threshold from Loss of GFR and Albuminuria" for publication in Toxics journal. The objective was to quantify changes in estimated GFR (eGFR) and albumin excretion (Ealb) according to levels of blood Cd ([Cd]b) and excretion of Cd (ECd) after adjustment for confounders among 482 residents of Cd-polluted and non-polluted regions of Thailand. ECd and Ealb were normalized to creatinine clearance (Ccr) as ECd/Ccr and Ealb/Ccr. Obtained results indicate that ECd/Ccr of 1.44 µg/L filtrate (0.01−0.02 µg/g creatinine) could determine a Cd threshold level from which protective exposure guidelines should be formulated. The idea is good. It should be tested through further investigation in another population even if the power of study is relatively strong. The authors were focused on the dose-response relationship, a critical step in the hazard assessment. Moreover, univariate analysis was used to quantify the independent effects of ECd/Ccr, eGFR, diabetes, and hypertension on Ealb/Ccr as well as blood Cd and eGFR as predictors of albuminuria. Based on the results, the authors are pointing to the observation that there are no safe levels of Cd exposure.
I suggest after proof reading the publication of a proposed manuscript.
RESPONSE: We thank the reviewer for her/his evaluation of our manuscript and for endorsing its scientific merit and public health significance.

Reviewer 3 Report
This manuscript reports on results that were obtained in the context of a study which was conducted with 482 residents from Thailand which aimed to gain insight into better defining a nephrotoxicity threshold for chronic cadmium exposure. It is suggested to avoid the abbreviation ‘GFR’ in the title (it could just be spelled out). Overall, the manuscript is well written, but the Materials and Methods section must be significantly improved as no details pertaining to the analysis of blood and urine for the total Cd concentrations were provided. Even though my expertise in statistics is rather limited, I understood the arguments that were made (they are coherent), but I cannot endorse them 100%. If some additional deficiencies are addressed (see my comments below), the manuscript can be re-evaluated for publication in Toxics.
Detailed comments:
Abstract
Line 18: It should read ‘burden of Cd leads to the progressive loss’.
Line 19: Please remove ‘in’ before according.
Line 30: It should read ‘These data indicate’.
Line 31: It should read ‘creatinine) may serve as a Cd threshold level based on which’.
Introduction
Line 48: It should read ‘in a Dutch prospective’.
Line 50: Please remove ‘in’.
Line 53: It is suggested to change ‘are limited’ to ‘is not well defined’.
Line 54: It should read ‘tubular’.
Line 60: It should read ‘death as a result of the cumulative burden of Cd’.
Line 61: Please remove ‘in’. It should read ‘cells complexed with’.
Materials and Methods
Line 73+74: ‘of filtrate’ is jargon and should be better explained for the lay reader (does this term refer to urine?).
Line 90: It should read ‘Cd concentration of the paddy’.
Line 92+93: it should read ‘survey revealed that the prevalence’.
Line 96+98: The ethics protocol numbers of the approved studies should be provided.
Line 108+109: ‘was based on ECd’ is cryptic. Is this referring to a one-time measurement of the urinary Cd concentration of any given person. Similarly does ‘[Cd]b’ refer to a one-time measurement of the blood Cd concentration? Please clarify.
Line 113+114: ‘measured by atomic absorption spectrometry’. Please provide details about the instrument that was used, whether it was flame or graphite furnace AAS and provide sufficient details (wavelength) for the reader to reproduce this analysis. If GFAAS was used were the blood samples digested (if so details about how this was accomplished need to be provided) or were matrix modifiers used in case GFAAS was used. Critical analytical details must be provide so the reader could reproduce this analyses.
Line 122: Details for the analytical methods that correspond to the ‘Urinary and plasma creatinine ([cr]u and [cr]p assays’ must be provided (e.g. references). ‘based on the Jaffe reaction ‘ is entirely unclear.
Line 123: A reference to the turbidimetric method that was used to measure albumin in urine must be provided.
Line 137: ‘Ex” is unclear and needs to be defined.
Line 148: It should read ‘to compare two groups’.
Line 161: It should read ‘of the Thai cohort participants, who were recruited’.
Table 1: The abbreviations ‘SBP’ and ‘DBP’ are unclear and must be explained.
Line 178: The abbreviation ‘ACR criterion’ is unclear and must be explained. DO the values ‘8.1%n and 15%’ refer to women or men or both? Please clarify.
Line b184: It should read ‘The corresponding mean’.
Line 206: It should read ‘investigate the dose’.
Line 224: It should read ‘who had low Cd’ (i.e. remove ‘of the’).
Line 258: The ‘prevalence odds ratio’ should be explained as it is unclear to readers not versed in math.
Line 278: It should read ‘An effect of Cd’.
Line 336: It should read ‘blood Cd levels’ to be consistent.
Line 353: Please remove ‘by’.
Line 360: It should read ‘In a subsequent analysis’.
Line 376: It should read ‘was observed between’.
Line 378: It should read ‘groups, the mean Ealb’.
Discussion
Line 386: It should read ‘tubular damage’.
Line 387: It should read ‘accumulation reduces the GFR’ and ‘and a decreasing GFR as a result.’.
Line 391: It should read analyzed data from’.
Line 398: Please remove ‘should it exist.’.
Line 408: It should read ‘weak and became’.
Line 428: It should read ‘albuminuria resulted’.
Line 443: It should read ‘The results described’.
Line 459: It should read ‘used also the’.
Line 464: It should read ‘exposure guidelines that provide sufficient health protection’.
Line 479: Please specify which agency provides the ‘5.24 μg/g creatinine’ threshold for Cd exposure.
Line 478-484: It is suggested to expand the conclusion slightly to reiterate the main findings. The formulation ‘threshold equivalent’ (line 483) could be better explained as I don’t think it was introduced/defined earlier in the manuscript.
I have provided some suggestions to improve the english style in my detailed comments.
Author Response
Reviewer 3
This manuscript reports on results that were obtained in the context of a study which was conducted with 482 residents from Thailand which aimed to gain insight into better defining a nephrotoxicity threshold for chronic cadmium exposure. It is suggested to avoid the abbreviation ‘GFR’ in the title (it could just be spelled out). Overall, the manuscript is well written, but the Materials and Methods section must be significantly improved as no details pertaining to the analysis of blood and urine for the total Cd concentrations were provided. Even though my expertise in statistics is rather limited, I understood the arguments that were made (they are coherent), but I cannot endorse them 100%. If some additional deficiencies are addressed (see my comments below), the manuscript can be re-evaluated for publication in Toxics.
RESPONSE: We thank this reviewer for their comments and suggestions to further improve our manuscript. GFR has now been spelt out in the title. Below we have provided point-by-point response to issues and concerns raised.
Detailed comments:
Abstract
Line 18: It should read ‘burden of Cd leads to the progressive loss’.
Line 19: Please remove ‘in’ before according.
Line 30: It should read ‘These data indicate’.
Line 31: It should read ‘creatinine) may serve as a Cd threshold level based on which’.
Introduction
Line 48: It should read ‘in a Dutch prospective’.
Line 50: Please remove ‘in’.
Line 53: It is suggested to change ‘are limited’ to ‘is not well defined’.
Line 54: It should read ‘tubular’.
Line 60: It should read ‘death as a result of the cumulative burden of Cd’.
Line 61: Please remove ‘in’. It should read ‘cells complexed with’.
Materials and Methods
Line 73+74: ‘of filtrate’ is jargon and should be better explained for the lay reader (does this term refer to urine?).
Line 90: It should read ‘Cd concentration of the paddy’.
Line 92+93: it should read ‘survey revealed that the prevalence’.
Line 96+98: The ethics protocol numbers of the approved studies should be provided.
Line 108+109: ‘was based on ECd’ is cryptic. Is this referring to a one-time measurement of the urinary Cd concentration of any given person. Similarly does ‘[Cd]b’ refer to a one-time measurement of the blood Cd concentration? Please clarify.
Line 113+114: ‘measured by atomic absorption spectrometry’. Please provide details about the instrument that was used, whether it was flame or graphite furnace AAS and provide sufficient details (wavelength) for the reader to reproduce this analysis. If GFAAS was used were the blood samples digested (if so details about how this was accomplished need to be provided) or were matrix modifiers used in case GFAAS was used. Critical analytical details must be provided so the reader could reproduce this analyses.
Line 122: Details for the analytical methods that correspond to the ‘Urinary and plasma creatinine ([cr]u and [cr]p assays’ must be provided (e.g. references). ‘based on the Jaffe reaction ‘ is entirely unclear.
Line 123: A reference to the turbidimetric method that was used to measure albumin in urine must be provided.
Line 137: ‘Ex” is unclear and needs to be defined.
Line 148: It should read ‘to compare two groups’.
Line 161: It should read ‘of the Thai cohort participants, who were recruited’.
Table 1: The abbreviations ‘SBP’ and ‘DBP’ are unclear and must be explained.
Line 178: The abbreviation ‘ACR criterion’ is unclear and must be explained. DO the values ‘8.1%n and 15%’ refer to women or men or both? Please clarify.
Line 184: It should read ‘The corresponding mean’.
Line 206: It should read ‘investigate the dose’.
Line 224: It should read ‘who had low Cd’ (i.e. remove ‘of the’).
Line 258: The ‘prevalence odds ratio’ should be explained as it is unclear to readers not versed in math.
Line 278: It should read ‘An effect of Cd’.
Line 336: It should read ‘blood Cd levels’ to be consistent.
Line 353: Please remove ‘by’.
Line 360: It should read ‘In a subsequent analysis’.
Line 376: It should read ‘was observed between’.
Line 378: It should read ‘groups, the mean Ealb’.
Discussion
Line 386: It should read ‘tubular damage’.
Line 387: It should read ‘accumulation reduces the GFR’ and ‘and a decreasing GFR as a result.’.
Line 391: It should read analyzed data from’.
Line 398: Please remove ‘should it exist.’.
Line 408: It should read ‘weak and became’.
Line 428: It should read ‘albuminuria resulted’.
Line 443: It should read ‘The results described’.
Line 459: It should read ‘used also the’.
Line 464: It should read ‘exposure guidelines that provide sufficient health protection’.
OVERALL RESPONSE TO TYPO ERRORS
- The typo errors in abstract, introduction, materials and methods, and discussion have all been corrected. Concerns other than typo errors are addressed below.
Concern 1: Line 73+74: ‘of filtrate’ is jargon and should be better explained for the lay reader (does this term refer to urine?).
- RESPONSE: The sentence has been changed to read as below.
“To depict an amount of Cd and albumin excreted per volume of filtrate, known also as primary urine, ECd and Ealb were normalized to Ccr as ECd/Ccr and Ealb/Ccr, respectively [23].”
Concern 2: Line 96+98: The ethics protocol numbers of the approved studies should be provided.
- RESPONSE: The ethics protocol numbers of the approved studies have been inserted (lines 97-100).
Concern 3: Line 108+109: ‘was based on ECd’ is cryptic. Is this referring to a one-time measurement of the urinary Cd concentration of any given person. Similarly does ‘[Cd]b’ refer to a one-time measurement of the blood Cd concentration? Please clarify.
Concern 4: Line 113+114: ‘measured by atomic absorption spectrometry’. Please provide details about the instrument that was used, whether it was flame or graphite furnace AAS and provide sufficient details (wavelength) for the reader to reproduce this analysis. If GFAAS was used were the blood samples digested (if so details about how this was accomplished need to be provided) or were matrix modifiers used in case GFAAS was used. Critical analytical details must be provided so the reader could reproduce these analyses.
Concern 5: Line 122: Details for the analytical methods that correspond to the ‘Urinary and plasma creatinine ([cr]u and [cr]p assays’ must be provided (e.g. references). ‘based on the Jaffe reaction ‘ is entirely unclear.
Concern 6: Line 123: A reference to the turbidimetric method that was used to measure albumin in urine must be provided.
RESPONSE TO CONCERNS 3-6
- We have rewritten Section 2.2. Assessment of Cadmium Exposure and Adverse Effects (lines) and references have been provided.
2.2. Assessment of Cadmium Exposure and Adverse Effects
Assessment of Cd exposure was based on one-time measurement of blood Cd concentration ([Cd]b) and the urinary excretion of Cd (ECd). Kidney functional assessment was based on Ealb and eGFR. For these measurements, samples of urine and whole blood were collected after overnight fast. Blood samples were collected within 3 h of urine collection. Aliquots of blood and urine samples were stored at −80°C for later analysis.
Levels of Cd in urine and blood ([Cd]u and [Cd]b) were quantified by graphite furnace atomic absorption spectrometry with the Zeeman-effect background correction system. Multielement standards (Merck KGaA, Darmstadt, Germany) were used for instrument calibration. The quality control and quality assurance of Cd quantitation were accomplished by simultaneous analysis of blood control samples (ClinChek, Munich, Germany) and the reference urine metal control levels 1, 2, and 3 (Lyphocheck, Bio-Rad, Hercules, CA, USA).
The limit of detection (LOD) for Cd in blood or urine, defined as 3 times the standard deviation of at least 10 blank sample measurements was 0.3 µg/L for [Cd]b and 0.1 µg/L for [Cd]u. The sample blanks, reference standards for urine and blood were included in the assay together with samples of urine or blood from subjects. Deionized water was used to zero an instrument. The coefficient of variation of Cd in the reference urine and blood samples were within acceptable clinical chemistry standards. When a sample contained Cd below its LOD, the Cd concentration assigned was the LOD value divided by the square root of 2 [31].
Urinary and plasma creatinine concentrations ([cr]u and [cr]p) were measured by the colorimetric method, based on the alkaline-picrate Jaffe’s reaction [32]. Urinary albumin concentration was determined by an immunoturbidimetric method [33, 34].
[32] Spencer, K. Analytical reviews in clinical biochemistry: The estimation of creatinine. Ann. Clin. Biochem. 1985, 23, 1-25.
[33] Bargnoux, A.S.; Barrot, A.; Fesler, P.; Kuster, N.; Badiou, S.; Dupuy, A.M.; Ribstein, J.; Cristol, J.P. Evaluation of five immunoturbidimetric assays for urinary albumin quantification and their impact on albuminuria categorization. Clin. Biochem. 2014, 47, 250-253.
[34] Kok, M.B.; Tegelaers, F.P.; van Dam, B.; van Rijn, J.L.; van Pelt, J. Carbamylation of albumin is a cause for discrepancies between albumin assays. Clin. Chim. Acta 2014, 434, 6-10.
Concern 7: Line 137: ‘Ex” is unclear and needs to be defined.
- RESPONSE: The definition of Ex has been defined as Excretion of x (line 146).
Concern 8: Line 178: The abbreviation ‘ACR criterion’ is unclear and must be explained. DO the values ‘8.1%n and 15%’ refer to women or men or both? Please clarify.
Concern 9: Line 479: Please specify which agency provides the ‘5.24 μg/g creatinine’ threshold for Cd exposure.
Concern 10: Line 478-484: It is suggested to expand the conclusion slightly to reiterate the main findings. The formulation ‘threshold equivalent’ (line 483) could be better explained as I don’t think it was introduced/defined earlier in the manuscript.
- RESPONSE TO CONCERNS 8-10: We have specified the authority that provide the referred threshold Cd exposure. We have rewritten part of the conclusion (lines 529-539), which now reads as below. The ‘threshold equivalent” has been change to read no-observed-adverse-effect level (NOAEL) equivalent which has been explained earlier in the text (lines 446-448).
“A dose-response relationship analysis of data from 482 non-occupationally exposed persons with a 1250-fold difference in Cd burden and a 667-fold difference in levels of blood Cd, environmental exposure to Cd was confirmed to be closely associated with a declining GFR and albuminuria. An independent effect of smoking on albuminuria has also been observed in smokers who had normal kidney function. The current World Health Organization Cd exposure limit of 5.24 µg/g creatinine (ECd/Ecr), which is solely based on an increment of excretion of β2-microglobulin above 300 µg/g creatinine underestimates the level at which Cd induces kidney damage. Our results show that when a declining GFR is considered along with albuminuria, the no-observed-adverse-effect level (NOAEL) equivalent is 0.01−0.02 µg/g creatinine. Now is the time to acknowledge there is no safe level of Cd exposure.”
Comments on the Quality of English Language
I have provided some suggestions to improve the English style in my detailed comments.
RESPONSE: All typo errors have been corrected and the English style has been adopted.
